# Regulatory Roles of Long Non-Coding RNAs Relevant to Antioxidant Enzymes and Immune Responses of *Apis cerana* Larvae Following *Ascosphaera apis* Invasion

**DOI:** 10.3390/ijms241814175

**Published:** 2023-09-16

**Authors:** Rui Guo, Siyi Wang, Sijia Guo, Xiaoxue Fan, He Zang, Xuze Gao, Xin Jing, Zhitan Liu, Zhihao Na, Peiyuan Zou, Dafu Chen

**Affiliations:** 1College of Animal Sciences (College of Bee Science), Fujian Agriculture and Forestry University, Fuzhou 350002, China; ruiguo@fafu.edu.cn (R.G.); siyiwang2021@163.com (S.W.); guosijia1998@163.com (S.G.); imfanxx@163.com (X.F.); zanghe321@163.com (H.Z.); gxz13845381765@163.com (X.G.); jingxin6662022@163.com (X.J.); 17339838732@163.com (Z.L.); n2176864946@163.com (Z.N.); zoupeiyuan2216@163.com (P.Z.); 2Apitherapy Research Institute, Fujian Agriculture and Forestry University, Fuzhou 350002, China

**Keywords:** non-coding RNA, long non-coding RNA, honey bee, *Apis cerana cerana*, *Ascosphaera apis*, antioxidant enzyme, immune response

## Abstract

Long non-coding RNAs (lncRNAs) play an essential part in controlling gene expression and a variety of biological processes such as immune defense and stress-response. However, whether and how lncRNAs regulate responses of *Apis cerana* larvae to *Ascosphaera apis* invasion has remained unclear until now. Here, the identification and structural analysis of lncRNAs in the guts of *A. cerana* worker larvae were conducted, and the expression profile of larval lncRNAs during the *A. apis* infection process was then analyzed, followed by an investigation of the regulatory roles of differentially expressed lncRNAs (DElncRNAs) in the host response. In total, 76 sense lncRNAs, 836 antisense lncRNAs, 184 intron lncRNAs, 362 bidirectional lncRNAs, and 2181 intron lncRNAs were discovered in the larval guts. Additionally, 30 known and 9 novel lncRNAs were potential precursors for 36 and 11 miRNAs, respectively. In the three comparison groups, 386, 351, and 272 DElncRNAs were respectively identified, indicating the change in the overall expression pattern of host lncRNAs following the *A. apis* invasion. Analysis of *cis*-acting effect showed that DElncRNAs in the 4-, 5-, and 6-day-old comparison groups putatively regulated 55, 30, and 20 up- and down-stream genes, respectively, which were involved in a series of crucial functional terms and pathways, such as MAPK signaling pathway, and cell process. Analysis showed that 31, 8, and 11 DElncRNAs as potential antisense lncRNAs may interact with 26, 8, and 9 sense-strand mRNAs. Moreover, investigation of the competing endogenous RNA (ceRNA) network indicated that 148, 283, and 257 DElncRNAs were putatively regulated. The expression of target genes by targeting corresponding DEmiRNAs included those associated with antioxidant enzymes and immune responses. These results suggested that DElncRNAs played a potential part in the larval guts responding to the *A. apis* infection through a *cis*-acting manner and ceRNA mechanisms. Our findings deepen our understanding of interactions between *A. cerana* larvae and *A. apis* and offer a basis for clarifying the DElncRNA-mediated mechanisms underlying the host response to fungal invasion.

## 1. Introduction

Long non-coding RNAs (lncRNAs) are a class of linear ncRNA molecules transcribed by RNA polymerase II or RNA polymerase III with a cap at the 5′ end and a tail at the 3′ end [1]. As compared with mRNAs, lncRNAs have some specific characteristics, such as low expression level, complex secondary structure, and low sequence conservation [2,3]. Accumulating evidence has shown that lncRNAs are involved in abundant biological processes, such as gene expression, chromatin modification, epigenetic regulation, and cell differentiation [2,4]. With the development of related knowledge and techniques, more and more lncRNAs were suggested to be able to translate into proteins or small peptides [4,5,6]. As a kind of versatile regulator, lncRNAs are capable of exerting functions through diverse manners, including the *cis*-acting effect and competing endogenous RNA (ceRNA) mechanism [7,8,9]. Currently, lncRNAs have been deeply studied in a few model organisms including human [5,10], mouse [11,12], and *Arabidopsis thaliana* [13]. However, progress associated with lncRNAs in invertebrates including insects is limited. Following transcriptomic investigation of lncRNAs in *Bombyx mori*, Wu et al. [14] documented that a subseries of lncRNAs putatively acted as ceRNAs or miRNA precursors to regulate biosynthesis, transport, and secretion of silk proteins. Xu et al. [15] reported that the *Bmdsx-AS*1 gene was highly expressed in the *B. mori* testis, and the splicing pattern of *Bmdsx* pre-mRNA was altered in the male silkworm after knock-down of *Bmdsx-AS*1. Jayakodi et al. [16] identified 2470 long intergenic non-coding RNAs (lincRNAs) and 1514 lincRNAs in the Asian honey bee (*Apis cerana*) and European honey bee (*Apis mellifera*), respectively, with 57% and 18% of lincRNAs showing tissue-dependent expression, and they discovered that 11 lincRNAs shared by *A. cerana* and *A. mellifera* were specifically regulated upon viral infection.

*Ascosphaera apis* is a lethal fungal pathogen that exclusively infects bee larvae and causes chalkbrood disease, resulting in severe losses for the apicultural industry [17]. Previous studies regarding chalkbrood are mainly relevant to *A. mellifera* larvae. For example, Chen et al. [18] detected that in the guts of *A. m. ligustca* worker larvae infected by *A. apis* 13 and 50 differentially expressed genes (DEGs) were involved in immune pathways, and the host cellular immune was extremely inhibited while the humoral immunity was induced to activation. In recent years, based on molecular and omics methods, our group conducted systematic investigation of responses of *A. cerana* worker larvae to *A. apis* invasion [19,20,21]; for example, Chen et al. [21] isolated *A. apis* from chalkbrood-like mummies of *A. c. cerana* drone larvae and verified that *A. apis* was able to infect worker larvae and caused chalkbrood disease; Du et al. [19] performed identification and expression profile analysis of miRNAs in the *A. c. cerana* worker 6-day-old larval guts infected by *A. apis* infection, followed by investigation of the potential regulatory roles of DEmiRNAs in the host response. Regarded as a major place from which insects excrete digestive enzymes and other enzymes, the midgut tissue is responsible for food digestion and nutrient absorption, defense against pathogenic microbes, and response to insecticides as well as toxins [22]. Increasing studies have demonstrated that lncRNAs are engaged in the responses of animals, including insects, to infections by various pathogens or parasites. For instance, Li et al. [23] suggested that lncRNAs may participate in *Enterovirus* 71 (*EV*71) infection-induced pathogenesis through regulating immune responses, protein binding, cellular component biogenesis and metabolism; Zhang et al. [24] observed that the CRISPR-Cas9-based knockout of the *VINR* gene in *Drosophila* cells enhanced the replication of the *Drosophila* C virus (DCV), and prevented the degradation of Cactin, which activated the host immune defense; Zhang et al. [25] found that inhibition of the expression of either MSTRG15394 or its target gene that encodes a secreted protease inhibitor (PI) promoted the accumulation of rice black-streaked dwarf virus (RBSDV) in the *Laodelphax striatellus* (Fallén) midgut. However, whether and how lncRNAs regulate the response of *A. cerana* larvae to *A. apis* invasion has remained unknown until the present.

In a previous study, the *A. c. cerana* worker 4-, 5-, and 6-day-old larval guts inoculated with *A. apis* spores and un-inoculated larval guts were prepared and subjected to deep sequencing [26]. Here, based on the gained transcriptome data, the identification and structural analysis of lncRNAs in the larval guts were conducted, and then the expression profile of lncRNAs in the larval guts following *A. apis* infection was analyzed, followed by an investigation of the regulatory roles of DElncRNAs in host response, with a focus on antioxidant enzymes and immune responses. Findings from this current work could not only lay a foundation for parsing the mechanism underlying the DElncRNA-modulated host response to *A. apis* invasion but also offer a novel insight into the interaction between *A. cerana* larvae and *A. apis*.

## 2. Results

### 2.1. Number, Type, and Property of lncRNAs in A. c. cerana Larval Guts

A total of 4032 *A. c. cerana* lncRNAs were discovered, including 3341 known lncRNAs and 691 novel ones (Figure 1A). Among these, there were 76 sense lncRNAs, 836 antisense lncRNAs, 184 intron lncRNAs, 362 bidirectional lncRNAs, and 2181 intron lncRNAs (Figure 1B). In addition, structural comparisons showed that the number of exons in lncRNAs ranged from 2 to 29, with the largest number of lncRNAs containing 2 exons (1264, 31.35%), while that of exons in mRNAs ranged from 1 to 133, with the largest number of mRNAs including 4 exons (1421, 13.19%); the quantity of introns in the lncRNAs ranged from 1 to 28, with the largest number of lncRNAs containing 1 intron (1264, 31.35%), whereas that of introns in mRNAs ranged from 0 to 132, with the largest number of mRNAs including 3 introns (1421, 13.19%) (Figure 1D); additionally, the lengths of exons, introns and transcripts in lncRNAs were shorter than those of mRNAs (Figure 1E–G). Moreover, as shown in Figure 1H, the overall expression level of lncRNAs was lower than that of mRNAs.

### 2.2. Analysis of DElncRNAs as Putative miRNA Precursors

Here, 30 known and 9 novel lncRNAs were predicted to be precursors for 36 and 11 miRNAs, respectively. Further analysis suggested that some lncRNAs may be precursors for several miRNAs; for example, XR_003698717.1 was the precursor for both ame-miR-252b and ame-miR-6057 (Figure 2A). Meanwhile, some miRNAs may be generated from the same precursor (lncRNA); for example, ame-miR-iab-4 was generated from the three lncRNAs including XR_003698299.1, XR_003698300.1, and XR_00369830.1 (Figure 2B).

### 2.3. Differential Expression Profile of lncRNAs in Larval Guts Following A. apis Infection

In the 4-, 5-, and 6-day-old comparison groups, 386, 351, and 272 DElncRNAs were identified (Figure 3A). Venn analysis revealed that 30 DElncRNAs were shared by these three comparison groups, while the numbers of specific DElncRNAs were 241, 177, and 141, respectively (Figure 3B). In addition, expression clustering analysis demonstrated that the 30 shared DElncRNAs presented various expression trends during the *A. apis* infection process (Figure 3C).

### 2.4. Cis-Acting Regulatory Manner of Host DElncRNAs

It is suggested that 68 DElncRNAs in the 4-day-old comparison group putatively regulated 55 up- and down-stream genes relevant to seven cellular component-associated GO terms including organelle and cell part, nine biological process-associated terms including single-organism process and biological regulation, and five molecular function-associated terms including catalytic activity and structural molecule activity (Figure 4A); these up- and down-stream genes were also involved in 69 KEGG pathways, such as aldosterone synthesis and secretin, the cGMP-PKG signaling pathway, phototransduction-*Drosophia*, the regulation of TRP pathways by inflammatory mediators, and cell gap junctions (Figure 4D). In the 5-day-old comparison group, 35 DElncRNAs potentially regulated 30 up- and down-stream genes relative to four cellular component-associated terms such as organelle and cell, six biological process-associated terms such as cellular process and regulation of biological process, and three molecular function-associated terms such as binding and catalytic activity (Figure 4B); these up- and down-stream genes were also engaged in 62 KEGG pathways including the metabolism pathway, aldosterone synthesis and secretin, and neurotrophin signaling pathway (Figure 4E). Additionally, 27 DElncRNAs in the 6-day-old comparison group putatively regulated 20 up- and down-stream genes, which were related to five cellular component-associated terms including membrane part and membrane, eight biological process-associated terms including response to stimulus and regulation of biological process, and four molecular function-associated terms including binding and signal transducer activity (Figure 4C). These up- and down-stream genes were also associated with two KEGG pathways, such as the stem cell pluripotency signaling pathway and the MAPK signaling pathway-*Drosophia* (Figure 4F). Further analysis indicated that DElncRNAs potentially modulated 16 genes relevant to a subseries of cellular or humoral immune pathways, such as endocytosis, Fc-γ-Receptor-Mediated Phagocytosis, and the MAPK signaling pathway (Table 1).

### 2.5. Investigation of Antisense DElncRNAs

It was detected that 13 known and 18 novel lncRNAs in the 4-day-old comparison group were potential antisense DElncRNAs, which may interact with 26 sense-strand mRNAs. For example, both MSTRG.7585.1 and MSTRG.7585.2 may interact with the ecdysone-inducible protein 75 isomer X2 (LOC107997697) gene (Figure 5A). In the 5-day-old comparison group, four known and four novel lncRNAs were putative antisense DElncRNAs, which may interact with eight sense-strand mRNAs. For example, MSTRG.7211.1 may interact with the Lactosamine 1,3-*N*-acetyl-β-D-glucosaminyltransferase (LOC107997326) gene (Figure 5B). In addition, two known and nine novel lncRNAs in the 6-day-old comparison group were predicted to be antisense DElncRNAs, which may interact with nine sense-strand mRNAs. For example, the novel lncRNA MSTRG.6380.3 may interact with the Class III Lipase (LOC107996467) gene, as shown in Figure 5C.

### 2.6. Analysis of DElncRNA-DEmiRNA Regulatory Networks

DElncRNA-DEmiRNA regulatory networks were constructed and analyzed, and the results demonstrated that 148 DElncRNAs in the 4-day-old comparison group could target 10 DEmiRNAs to form a complex regulatory network (Appendix A). Most of these DElncRNAs (98, 66.22%) were found to target only one DEmiRNA, while some other DElncRNAs (50, 33.78%) had two or three target DEmiRNAs. For example, XR_003697465.1 could target three DEmiRNAs including miR-2770-y, miR-4638-y, and miR-4968-y (Appendix A). Comparatively, 283 DElncRNAs in the 5-day-old comparison group could target 27 DEmiRNAs (Appendix A); among these DElncRNAs, 177 (62.54%) were found to target several DEmiRNAs, whereas 106 (37.46%) DElncRNAs had only one DEmiRNA (Appendix A). In the 6-day-old comparison group, 257 DElncRNAs could target 54 DEmiRNAs (Appendix A), and the majority of these DElncRNAs (206, 80.16%) were observed to target several DEmiRNAs, while 51 (19.84%) DElncRNAs could target only one DEmiRNA (Appendix A).

### 2.7. Antioxidant Enzyme-Associated ceRNA Sub-Network of Host DElncRNAs

Further analyses suggested that 56 DElncRNAs could target five DEmiRNAs and further target three DEmRNAs shared by the three comparison groups, forming a complex regulatory network relevant to antioxidant enzymes (Figure 6). In detail, nine DElncRNAs could target miR-1277-x, further targeting one DEmRNA encoding superoxide dismutases (SOD); 34 DElncRNAs could target two miRNAs (miR-1344-x and novel-m0032-5p) and further target one DEmRNA encoding catalases (CAT); 13 DElncRNAs could target two DEmiRNAs (novel-m0044-5p and novel-m0003-3p) and further bind to one glutathione S-transferase (GST)-encoding DEmRNA.

Diamonds represent miRNAs, circles represent lncRNAs, triangles represent antioxidant enzymes, and hexagons represent mRNAs.

### 2.8. Immune Response-Associated ceRNA Sub-Network of Host DElncRNAs

CeRNA sub-networks associated with cellular and humoral immune response were also investigated. The results showed that 213 (263) DElncRNAs in the 4-day-old comparison group putatively targeted 12 (37) miRNAs, further linking to two (four) mRNAs (Figure 7); 204 (234) DElncRNAs in the 5-day-old comparison group putatively targeted 12 (37) miRNAs, further linking to seven (ten) mRNAs (Appendix A); and 40 (186) DElncRNAs in the 6-day-old comparison group putatively targeted 12 (37) miRNAs, further linking to two (four) mRNAs (Appendix A).

### 2.9. RT-qPCR Detection of DElncRNAs

The RT-qPCR results suggested that the expression trends of five DElncRNAs randomly chosen from the three comparison groups were in accordance with those seen in the transcriptome data (Figure 8), which confirmed the reliability of the sequencing data used in this work.

## 3. Discussion

### 3.1. A. c. cerana lncRNAs Shared Analogous Structural Property with Those in Other Animals, Plants, and Microorganisms

Here, 4032 lncRNAs were for the first time identified in the guts of *A. c. cerana* worker larvae, including 3341 known and 691 novel lncRNAs (Figure 1A). The identified lncRNAs could enrich the reservoir of *A. cerana* lncRNAs, offering a valuable resource for related studies. Considering the tissue-specific expression of lncRNAs, the discovered lncRNAs in this work were inferred to be only a fraction of the total lncRNAs in *A. c. cerana*. It is believed that more lncRNAs will be detected based on more transcriptome datasets derived from more tissues or organs. Structural analysis showed that in comparison with the mRNAs, *A. c. cerana* lncRNAs had fewer and shorter exons and introns, shorter transcript lengths, and lower expression levels, analogous to those lncRNAs discovered in the *A. c. cerana* workers’ midguts [27]. Also, these structural features were shared by lncRNAs identified in humans [28], *Plutella xylostella* [29], *Oryza sativa* [30], and *Nosema ceranae* [31]. These results suggested that the structural features of lncRNAs were conserved in animals, plants, and microorganisms.

### 3.2. A. apis Invasion Gave Rise to the Alteration of Overall Expression Pattern of lncRNAs in the Larval Guts

Following the *A. apis* infection, 386, 351, and 272 DElncRNAs were detected in the 4-, 5-, and 6-day-old comparison groups, respectively, indicative of the alteration of the overall expression pattern of host lncRNAs caused by fungal invasion. This is similar to findings from other insects, such as *Sarcophaga peregrina*, *Aedes aegypti*, and *B. mori* in response to various pathogens or parasites [32,33,34]. In addition, 30 DElncRNAs (MSTRG. 10318.5, MSTRG. 9797.1, and XR_001765563.2 etc.) were shared by the above-mentioned three comparison groups (Figure 3C). These 30 DElncRNAs were speculated to exert regulatory functions of great importance during the *A. apis* response of *A. c. cerana* worker larvae, thus deserving further investigation in the near future.

### 3.3. A. c. cerana lncRNAs May Be a Source for the Generation of Abundant miRNAs

An increasing number of studies have indicated that lncRNAs are able to act as miRNA precursors to generate a large quantity of mature miRNAs through an array of splicing and catalysis processes [35]. However, knowledge of lncRNAs as miRNA precursors in insects, including honey bees, is still limited at present. Based on the sequencing of *Metarhizium anisopliae*-infected and un-infected *Plutella xylostella* fat body tissues, Zafar et al. [36] found that multiple lncRNAs acting as miRNA precursors potentially regulated the expression of mRNAs involved in immunity and development. In this work, 30 known and 9 novel lncRNAs were predicted to be putative precursors for 36 and 11 miRNAs, respectively. Additionally, we found that the same lncRNAs could be precursors for several different miRNAs (Figure 2A), while several different lncRNAs as precursors could produce the same one miRNA (Figure 2B). This phenomenon was also observed in other species, such as *Capra hircus* and *Cucumis melo* L. [37,38]. Collectively, it is inferred that there is an intrinsic relationship between lncRNAs and miRNAs in vertebrates, and lncRNAs may be another primary resource for generating abundant miRNAs. The MAPK signaling pathway is responsible for the oxidative stress response and also acts as a signaling hub that can be activated by external signals [39]. In insects, a series of signaling pathways, including MAPK, Ras, mTOR, and PI3K/Akt signaling pathways, jointly modulate numerous biological processes, such as developmental timing and organ growth [40,41]. Here, XR_003698301.1 was detected to generate ame-miR-iab-4, which putatively targeted the mRNA (LOC107996893) of the mitogen-activated protein kinase gene. This implies that XR_003698301.1 potentially participated in the larval response to *A. apis* invasion by generating ame-miR-iab-4.

### 3.4. DElncRNAs Potentially Modulated Host Response to A. apis Invasion by Regulating the Transcription of Up- and Down-Stream Genes 

Accumulating evidence has shown that lncRNAs are capable of regulating the transcription of up- and down-stream genes in a *cis*-acting or *trans*-acting manner and promote life progression processes such as cell proliferation, apoptosis, and immune response [2,4]. Here, *cis*-acting effect analysis suggested that 55 and 20 up- and down-stream genes of DElncRNAs in the 4- and 6-day-old comparison groups were involved in immune system progress-relevant functional terms such as response to stimulus (GO: 0050896) and signaling (GO: 0023052); these up- and down-stream genes were also engaged in pathways such as the MAPK signaling pathway (KO: 04010) and TNF signaling pathway (KO: 04668) (Figure 4). The results suggested that corresponding DElncRNAs may participate in the host immune response to *A. apis* infection. Invasions by pathogenic microorganisms activate not only cellular immunity including endocytosis and autophagy but also humoral immunity such as the Jak/STAT, Toll and Imd signaling pathways [42]. The Toll and Imd signaling pathways in insects play critical roles in regulating humoral immune responses; however, related studies were mainly focused on *Drosophila melanogaster*. Li et al. [43] verified that Toll activates JNK-mediated apoptosis through ROS production and further affected organ development and tissue homeostasis of *Drosophila*. The Toll signaling pathway involves prevention of autoimmunity and recognition receptors; a serine protease cascade results in the cleavage of pro-Spaetzle into mature Spaetzle, which binds to the membrane-anchored and induces the synthesis of antimicrobial peptides (AMPs) [42]. Here, two DElncRNAs, including MSTRG.10453.1 and XR_003698198.1, were found to regulate the transcription of the apidaecins gene (log_2_FC = 2.50), indicating that these two DElncRNAs were likely to be engaged in the host response to *A. apis* infection by activating the Toll and IMD signaling pathways to induce the synthesis and release of AMPs.

### 3.5. Antisense DElncRNAs May Participate in the Response of Larvae to A. apis Invasion by Interacting with Corresponding Sense-Strand mRNAs

The action modes of lncRNAs are flexible and diverse; antisense lncRNAs are transcribed from the antisense strand of protein-coding genes, whose sequences overlap with the genes’ sense strands and can reverse-bind with mRNA to form RNA-RNA dimeric structures, thereby regulating gene silencing and transcription, as well as mRNA stability [4]. Guo et al. [44] proved that the activation and regulation of the MAPK cascade is influenced by the intercommunication between two major insect hormones, 20-hydroxyecdysone (20E) and juvenile hormone (JH). This intricate interaction plays a crucial role in triggering significant physiological responses, such as growth and development, reproduction, or immune defense function. This causes a switch to *trans*-regulate the differential expression of aminopeptidase N and other genes in *Plutella xylostella* midgut tissue, countering the virulence effect of *Bacillus thuringiensis* (Bt) toxins. In this current work, MSTRG.7585.2 (log_2_FC = −1.31) in the 4-day-old comparison group was observed to bind to the sense-strand mRNA of the ecdysone-inducible protein 75 isomer X2 (LOC107997697) gene (Figure 5A), indicating that MSTRG.7585.2 may participate in the host immune response to *A. apis* invasion by acting as an antisense lncRNA. The insulin-like peptide receptor, a modulator of the balance between reproduction and immunity in insects, is present in all multicellular organisms. In *Drosophila*, it participates in the regulation of diverse processes such as growth, development, metabolic homeostasis, and lifespan [45,46]. McCormack et al. [47] reported that inhibition of the insulin/insulin-like growth factor signaling pathway prolonged the lifespan of *Drosophila*, and mutations in the insulin receptor substrate Chico extended the survival of mutant flies by enhancing the host defense against infections by various bacterial pathogens. Here, MSTRG.12965.2 (log_2_FC = −1.19) in the 4-day-old comparison group was putatively interacted with the sense-strand mRNA of the insulin-like peptide receptor (LOC108002698) gene (Figure 5A), which suggested that MSTRG.12965.2 as an antisense lncRNA may modulate the transcription of the insulin-like peptide receptor gene and further participate in the host immune response to *A. apis* invasion.

### 3.6. DElncRNAs Putatively Participated in the Regulation of Larval A. apis—Response through Modulating the Expression of Target Genes via ceRNA Networks

MiRNAs play a crucial role in gene expression and numerous pivotal biological processes such as growth, development, behavioral change, signal transduction, and immune response [48,49]. Wang et al. [48] detected that *Drosophila* adult mutants lacking miR-124 showed severe problems in locomotion, flight, and female fertility. Yuan et al. [49] reported that miR-315 inhibited the expression of the target gene encoding the fragile X mental retardation protein (FMRP), which further impacted the structure and function of *Drosophila* neurosynapses; knockdown of miR-315 led to embryonic death while the overexpression of miR-315 gave rise to pupation defects and reduced hatching rates. LncRNAs containing miRNA response elements (MREs) have been suggested to absorb miRNAs as “molecular sponges”, thereby reducing the binding of miRNAs to target mRNAs and attenuating the repressive effects on target genes [50,51]. Jiang et al. [52] identified a lncRNA named lncRNA23468 that contained conserved eTM (endogenous target mimics) sites for miR482b in tomato and found that the expression level of miR482b was significantly decreased while that of the target genes *NBS-LRRs* was significantly increased after the lncRNA23468 overexpression, resulting in enhanced resistance to *Phytophthora infestans*, which suggested that lncRNA23468 was capable of regulating host resistance via the ceRNA mechanism. Reactive oxygen species (ROS) are byproducts of steroid hormone production. There are two antioxidant systems containing enzymatic and non-enzymatic antioxidants that keep the amount of ROS in balance, which can modulate viability, activation, and proliferation of cells as well as organ function [53]. Braun et al. [54] documented that the SOD2 was crucial for physiological persistence of *Corpora lutea* and also acted as a signaling molecule securing the *C. lutea* from apoptosis and insuring long-term luteal cell survival. Li et al. [55] observed that the enzymatic activities of superoxide dismutase, catalase, and glutathione S-transferase were significantly elevated in the guts of uninfected larvae in comparison with those in the *A. apis*-infected larval guts, indicating that the *A. apis* infection may compromise the ability of the *A. apis*-infected larvae to cope with oxidative stress. The increase in SOD and CAT enzymes was closely associated with extended lifespan and the upregulation of cellular defenses against oxidative damage in *D. melanogaster* [56,57]. Here, based on investigation of the antioxidant enzyme-associated ceRNA regulatory network, we found that nine DElncRNAs (MSTRG.5378.3, MSTRG.13164.1, MSTRG.2452.1 etc.) could target one DEmiRNA, which further targets one DEmRNA encoding SOD; whereas 34 (MSTRG.11735.1, MSTRG.1386.1, MSTRG.678.1 etc.) and 13 (MSTRG.1784.1, MSTRG.4471.3, XR_003698051.1 etc.) DElncRNAs could target two DEmiRNAs, and two DEmiRNAs and further target one CAT-encoding DEmRNA and GST-encoding DEmRNA (Figure 6). These results demonstrated that corresponding DElnRNAs were likely to be associated with antioxidant enzymes that are potentially involved in regulating the immune response of worker bee larvae to *A. apis* infestation.

Intracellular protein degradation is involved in life activities such as cell growth and differentiation, response to infections by pathogenic microorganisms, and apoptotic cell death modulations through the ubiquitin protease system and autophagic pathway [58]. In eukaryotes, intracellular protein degradation involves the E1 ubiquitin-activating enzyme, the E2 ubiquitin-conjugating enzyme, and the E3 ubiquitin-ligating enzyme, which function as critical regulators of many cellular processes including cell cycle, division, and apoptosis [59,60]. Choi et al. [61] observed that silencing an E3 ubiquitin ligase, a linear ubiquitin chain assembly complex (LUBAC), was capable of increasing DNA double-strand breaks (DSBs) in old fly brain and inducing apoptosis and neurodegeneration in the Alzheimer disease (AD) model fly brain. Kanoh et al. [62] found that the small ubiquitin-like modifier (SUMO) and E3 ubiquitin-ligating enzyme mediated the humoral immune pathway-Toll signaling pathway in *Drosophila*. Here, 112, 124, and 95 DElncRNAs were detected to target ten miRNAs, which may regulate the expression of the ubiquitin-conjugating enzyme E2 gene Q2 (LOC108002565) by targeting corresponding mRNAs (Appendix A). In addition, the downregulation of both XR_001765563.2 and ubiquitin-binding enzyme E2 gene Q2 were observed in the 5- and the 6-day-old comparison groups; while miR-250-x, a DEmiRNA interacting with both XR_001765563.2 and ubiquitin-binding enzyme E2 gene Q2, was up-regulated. Together, the results revealed that XR_001765563.2 was likely to modulate the expression of the ubiquitin-conjugating enzyme E2 gene Q2 and downstream processes using miR-250-x as a bridge. The XR_001765563.2/miR-250-x/ubiquitin-conjugating enzyme E2 gene Q2 axis may play an essential part in host response to *A. apis* invasion.

## 4. Materials and Methods

### 4.1. Fungi and Bee Larvae

*A. apis* was previously isolated and kept at the Honey Bee Protection Laboratory of the College of Animal Sciences (College of Bee Science), Fujian Agriculture and Forestry University, Fuzhou, China [18,26]. Shortly, one chalkbrood mummy of *A. c. cerana* was subjected to surface sterilization with 10% sodium hypochlorite for 10 min, and then washed utilizing sterile water for 2 min; the rinsed mummy was cut up and then placed on sterile Petri plates with potato-dextrose agar (PDA) containing broad-spectrum antibiotics (50 ppm of streptomycin and ampicillin); seven days later, the pure culture of *A. apis* on the PDA plates were conserved at 4 °C.

*A. c. cerana* worker larvae were derived from three colonies reared in the apiary of the College of Animal Sciences (College of Bee Science) (119.2369° E, 26.08279° N), Fujian Agriculture and Forestry University, Fuzhou, China. These three *A. c. cerana* colonies had strong populations and were free of infections by two widespread microsporidia (*Nosema ceranae* and *Nosema apis*) and seven common viruses (SBV, KBV, ABPV, BQCV, CBPV, IAPV, and DWV) [63].

### 4.2. Transcriptome Data Source

In a previous study, by using a transferring spoon, the *A. c. cerana* worker 2-day-old larvae were transferred from three colonies to 6-well culture plates; the 3-day-old larvae (*n* = 48) in the treatment group were fed an artificial diet containing spores of *A. apis* with the final concentration of 10^7^ spores/mL, whereas the 3-day-old larvae (*n* = 48) in the control group were fed an artificial diet without spores of *A. apis*; the larvae in the treatment group and control group were respectively reared in two incubators (35 ± 0.5 °C, RH 90%); three gut tissues of 4-, 5-, and 6-day-old larvae in the treatment group (AcT1, AcT2, and AcT3) and those in the control group (AcCK1, AcCK2, and AcCK3) were respectively dissected using the method described by our group [20], followed by RNA extraction and cDNA library construction, and ultimately, the six cDNA libraries were sequenced on an Illumina HiSeq 4000 platform [26]. The produced raw datasets are available in the NCBI Short Read Archive (SRA) database (http://www.ncbi.nlm.nih.gov/sra/ (accessed on 10 March 2023)) under BioProject number: PRJNA565611.

### 4.3. Quality Control and Genomic Mapping

According to the method described by Chen et al. [18], the Fastp software (version 0.18.0) [64] was employed to perform the quality control of the generated raw reads, which were filtered by removing reads containing adapters, unknown nucleotides (N) (constituting more than 10%), and low quality (*Q*-value ≤ 20) bases (constituting more than 50%) to gain high-quality clean reads.

By using the Bowtie 2 software (version 2.2.8) [65], the clean reads were mapped to the ribosome RNA (rRNA) database (https://www.arb-silva.de/ (accessed on 10 March 2023)). After removing the mapped clean reads, the unmapped clean reads were then mapped to the reference genome of *A. cerana* (assembly ACSNU-2.0) utilizing HISAT2 software (accessed on 10 March 2023) [66], and the mapped clean reads were used for subsequent analyses.

### 4.4. Identification of lncRNAs

The identification of lncRNAs was conducted following our previously described protocol [31]: (1) the transcriptome was assembled using the StringTie (http://ccb.jhu.edu/software/stringtie/index.shtml (accessed on 10 March 2023)) based on the clean reads mapped to the reference genome; (2) the transcripts with lengths more than 200 nt and have more than two exons were selected as known lncRNA candidates; (3) the CNCI (version 3.0) [67] and CPC (version 2.0) [68] software were respectively utilized in combination to sort novel lncRNA candidates from putative protein-coding RNAs in the unknown transcripts by default parameters, and the intersection of both sets of results was regarded as comprising the novel lncRNAs with high confidence.

### 4.5. Analysis of lncRNAs as Potential miRNA Precursors

LncRNAs have been shown to serve as precursors to generate multiple mature miRNAs [14]. Here, (1) potential miRNA precursors were predicted by aligning the sequences of lncRNAs to the miRBase database (http://www.mirbase.org/), (accessed on 11 March 2023) with the Blast tool, and only those with comparison coverage >90% were selected; (2) miRNAs and their precursors derived from DElncRNAs were identified using the miRPara software (version 6.3) [69]; (3) the intersecting set was regarded as the highly confident miRNA precursors.

### 4.6. Differential Analysis of lncRNAs

By using the edgeR software (http://www.bioconductor.org/ version 4.2) (accessed on 11 March 2023) [70], the DElncRNAs in the AcCK1 vs. AcT1, AcCK2 vs. AcT2, and AcCK3 vs. AcT3 comparison groups were screened following the criteria of *p* ≤ 0.05 (corrected by false discovery rate) and |log_2_(FC)| ≥ 1. Subsequently, the Venn diagram analysis and expression clustering of the DElncRNAs were carried out with corresponding tools in the OmicShare platform (https://www.omicshare.com/tools/Home/Soft/venn (accessed on 11 March 2023)).

### 4.7. Investigation of the Cis-Acting Effect of DElncRNAs

It has been shown that an emerging class of lncRNAs control the expression of neighboring protein-coding genes via a *cis*-acting effect [71]. On the basis of the genomic colocalizations of DElncRNAs, protein-coding genes located 10 kb up- and down-stream of DElncRNAs were surveyed, which were then annotated to the GO (http://www.geneontology.org/ (accessed on 11 March 2023)) and KEGG (https://www.kegg.jp/ (accessed on 11 March 2023)) databases by the Blast tool, respectively, to yield functional terms and pathways.

### 4.8. Investigation of the Trans-Acting Effect of DElncRNAs

Antisense lncRNAs have been suggested to interact with sense-strand mRNAs and exert their function by regulating the abundance of target mRNAs [72]. Here, the RNAplex software (version 2.6.3) [73] was employed to predict the complementary pairing relationships between antisense lncRNAs and mRNAs in each comparison group, following the criteria of a thermodynamic structure with free energy.

### 4.9. Construction and Analysis of ceRNA Networks of DElncRNAs

Recent studies have indicated that lncRNAs are capable of absorbing target miRNAs and indirectly regulating downstream gene expression via ceRNA networks [74,75]. The potential targeting relationships between immune-defense-related DElncRNA and miRNAs, as well as miRNAs and mRNA, were predicted with RNAhybrid (V2.1.2) [76], MiRanda (V3.3a) [77], and TargetScan software (version 8.0) [78]. On the basis of the predicted targeting relationships, the DElncRNA-DEmiRNA and DELncRNA-miRNA-mRNA regulatory networks were constructed and then visualized by using Cytoscape v.3.6.1 software [79].

### 4.10. Investigation of Antioxidant Enzyme- and Immune Response-Associated DElncRNAs and the Corresponding Regulatory Network

Following the method described by Guo et al. [80], the target DEmiRNAs of DElncRNAs and the target DEmRNAs of DElncRNA-targeted DEmiRNAs were respectively predicted utilizing RNAhybrid in combination with MiRanda and TargetScan. Next, based on the predicted targeting relationships, the DElncRNA-DEmiRNA-DEmRNA regulatory networks were constructed. Further, based on the annotations in both the Nr and KEGG databases, target DEmRNAs and corresponding DEmiRNAs as well as DElncRNAs associated with antioxidant enzymes and immune responses were respectively selected for the construction of sub-networks, followed by visualization with Cytoscape v3.6.1 software.

### 4.11. RT-qPCR Validation of DElnRNAs

To verify the reliability of the transcriptome data used in this work, five DElncRNAs from each of three comparison groups mentioned above were randomly chosen for RT-qPCR detection. The five DElncRNAs from the AcCK1 vs. AcT1 comparison groups were XR_003697465.1, XR_001765834.2, XR_003697582.1, MSTRG.9797.1, and MSTRG.1072.5; the five DElncRNAs from the AcCK2 vs. AcT2 comparison group were XR_003696857.1, XR_001765043.2, XR_003696191.1, XR_001765563.2, and XR_001766844.2; the five DElncRNAs from the AcCK3 vs. AcT3 comparison groups were XR_003697158.1, XR_003697274.1, XR_003696193.1, XR_003697582.1, and MSTRG.13245.9. Specific primers (Appendix A) for these 15 DElncRNAs were designed with Primer Premier 6 [81], and then synthesized by Sangon Biotech (Shanghai) Co., Ltd., Shanghai, China. The *actin* gene (GenBank accession number: XM_017059068.2) was used as an internal reference. Total RNA from the gut tissues of *A. apis*-inoculated and un-inoculated 4-, 5-, and 6-day-old larvae (*n* = 3) was respectively isolated using the RNA extraction kit (Promega, Madison, WI, USA). The resulting cDNAs (Yeasen, Shanghai, China) were used as templates for qPCR reactions, which were performed under the following conditions: 95 °C pre-denaturation for 3 min; 95 °C denaturation for 15 s, 56 °C annealing for 30 s and extension for 20 s, for a total of 40 cycles. The reaction system (20 μL) contained 1 μL of cDNA, 10 μL of qPCR, 1 μL of upstream and downstream primers (2.5 μmol/L), and 7 μL of DEPC water. Each reaction was performed with at least three samples in parallel and was repeated three times. The relative expression level of each DElncRNA was calculated by the 2^−∆∆CT^ method [82]. GraphPad Prism 8.0 software was used for data analysis and plotting. Data are shown as mean ± standard deviation (SD) and were subjected to Student’s *t*-test, ns: *p* > 0.05, *: *p* < 0.05, **: *p* < 0.01, ***: *p* < 0.001, ****: *p* < 0.0001).

## 5. Conclusions

In conclusion, 3341 known and 691 novel lncRNAs were identified in the *A. c. cerana* worker larval guts, and the structural characteristics of these lncRNAs were analogous to those discovered in other animals, plants, and microorganisms; 39 lncRNAs were potential precursors for miRNAs. The overall expression pattern of host lncRNAs was altered owing to the *A. apis* invasion. Corresponding DElncRNAs were putative regulators in the larval response to the *A. apis* infection through diverse mechanisms, such as the direct regulation of the transcription of up- and down-stream genes, interaction with sense-strand mRNAs as antisense lncRNAs, and the indirect modulation of target gene expression via ceRNA networks including sub-networks associated with antioxidant enzymes and immune responses. The XR_001765563.2/miR-250-x/E2 ubiquitin-conjugating enzyme gene Q2 axis was likely to play a critical part in the host *A. apis*-response (Figure 9). These findings resolved the regulatory manners and potential roles of DElncRNAs in the response of *A. c. cerana* worker larvae to *A. apis* infection, laid a foundation for clarifying the molecular mechanisms underlying the host response, and provided candidates for further functional dissection and future chalkbrood diagnosis and control.

## Figures and Tables

**Figure 1 ijms-24-14175-f001:**
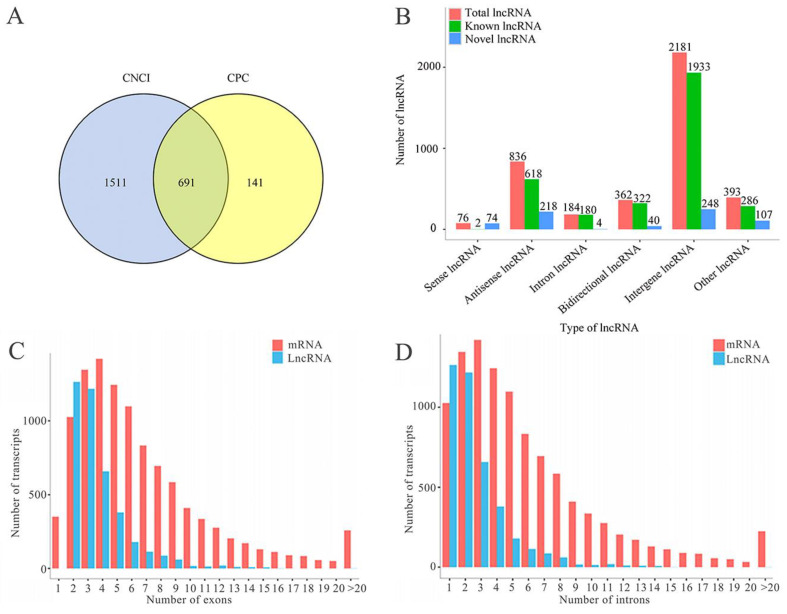
Identification and structural characterization of lncRNAs in the *A. c. cerana* workers’ larval guts. (**A**) Venn analysis of lncRNAs predicted by CNCI and CPC software; (**B**) Type statistics of lncRNAs; (**C**) Comparison of exon number between lncRNAs and mRNAs; (**D**) Comparison of intron number between lncRNAs and mRNAs; (**E**) Comparison of exon length between lncRNAs and mRNAs; (**F**) Comparison of intron length between lncRNAs and mRNAs; (**G**) Comparison of transcript length between lncRNAs and mRNAs; (**H**) Comparison of expression level between lncRNAs and mRNAs.

**Figure 2 ijms-24-14175-f002:**
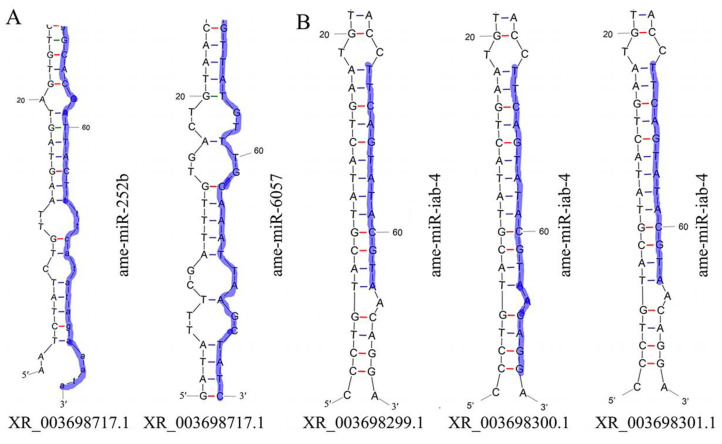
Stem–loop structures of lncRNAs served as potential miRNA precursors. (**A**) XR_003698717.1, the potential precursor for both ame-mir-252b and ame-mir-6057; (**B**) ame-miR-iab-4, the potential generated from XR_003698299.1, XR_003698300.1, and XR_00369830.1. The nucleotides in purple represent the mature sequences of these five miRNAs.

**Figure 3 ijms-24-14175-f003:**
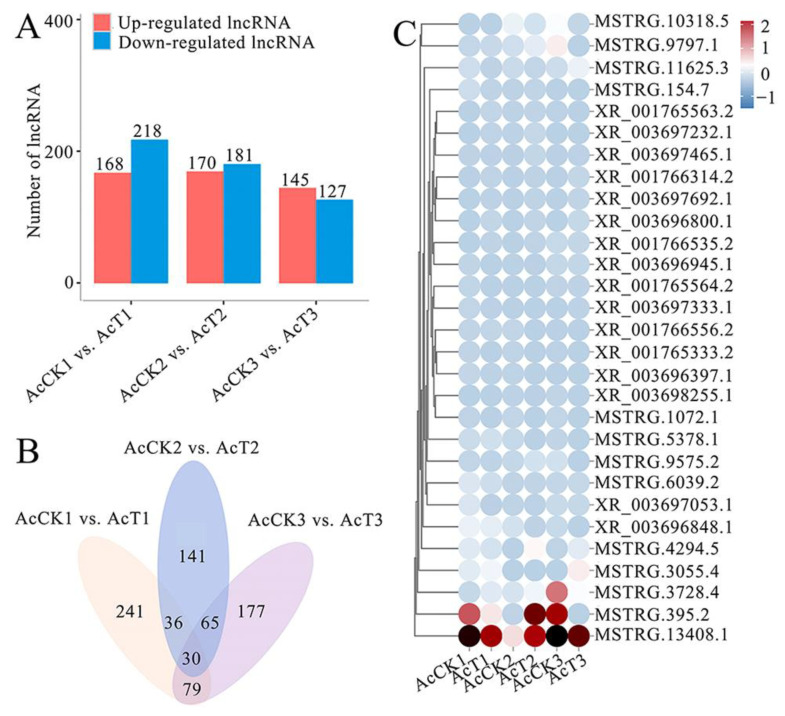
Differential analysis of lncRNAs in the larval guts of *A. c. cerana* workers infected by *A. apis*. (**A**) Quantitative statistics of DElncRNAs in three comparison groups; (**B**) Venn analysis of DElncRNAs in three comparison groups; (**C**) Expression clustering of the DElncRNAs shared by three comparison groups.

**Figure 4 ijms-24-14175-f004:**
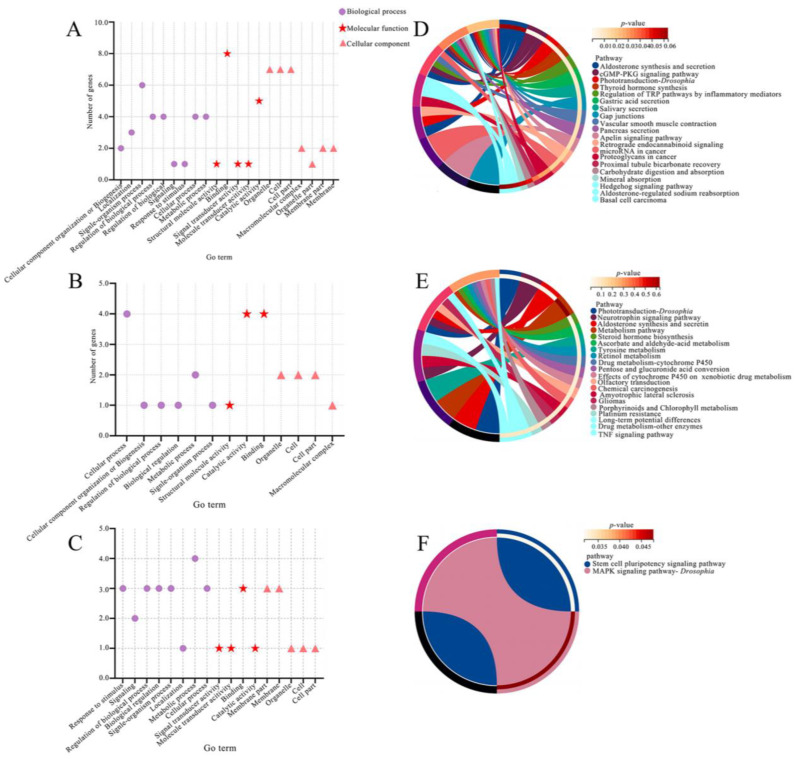
Functional terms and pathways enriched by up- and down-stream genes of DElncRNAs. (**A**,**C**,**E**) GO terms annotated by up- and down-stream genes of DElncRNAs in three comparison groups; (**B**,**D**,**F**) KEGG pathways enriched by up- and down-stream genes of DElncRNAs in three comparison groups.

**Figure 5 ijms-24-14175-f005:**
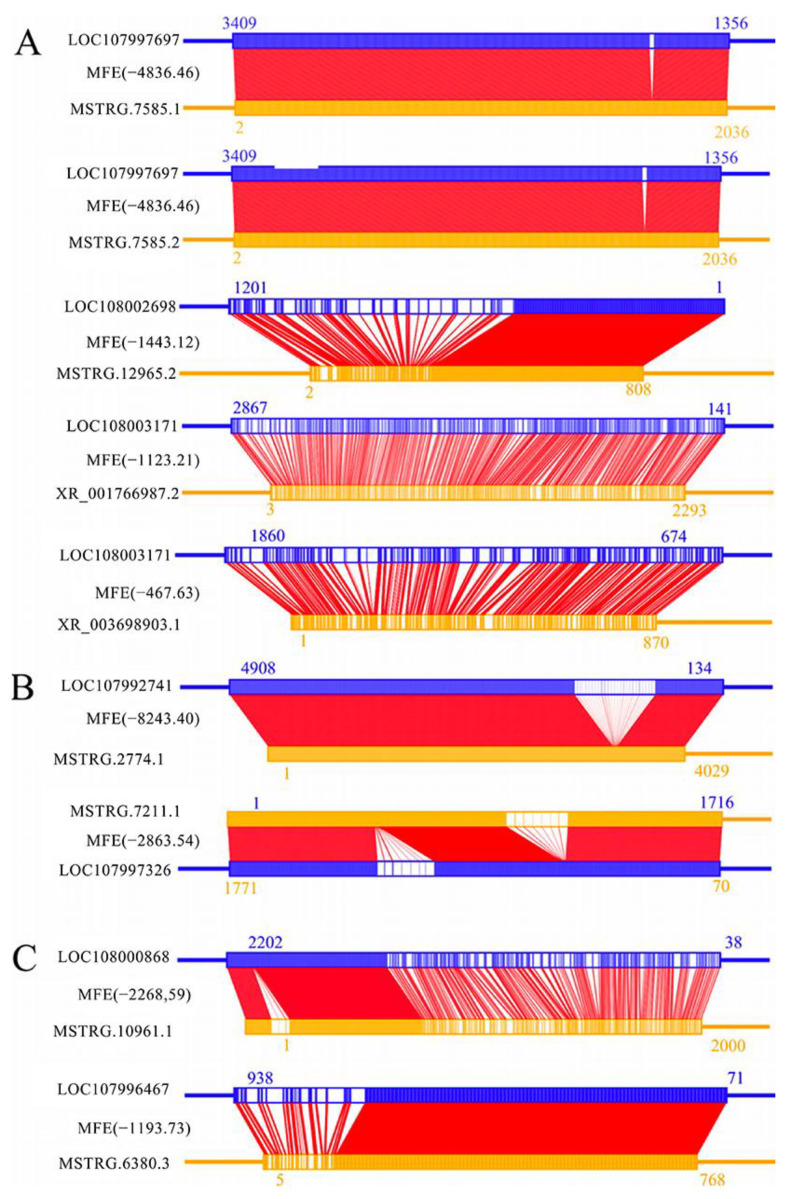
Binding relationships between nine antisense DElncRNAs and corresponding sense-strand mRNAs. (**A**) Two known (XR_001766987.2, XR_003698903.1) and three novel lncRNAs (MSTRG.7585.1, MSTRG.7585.2, MSTRG.12965.2) as potential antisense lncRNAs in the 4-day-old comparison group; (**B**) Two novel lncRNAs (MSTRG.2774.1, MSTRG.7211.1) as potential antisense lncRNAs in the 5-day-old comparison group; (**C**) Two novel lncRNAs (MSTRG.10961.1, MSTRG.10961.2) as potential antisense lncRNAs in the 6-day-old comparison group.

**Figure 6 ijms-24-14175-f006:**
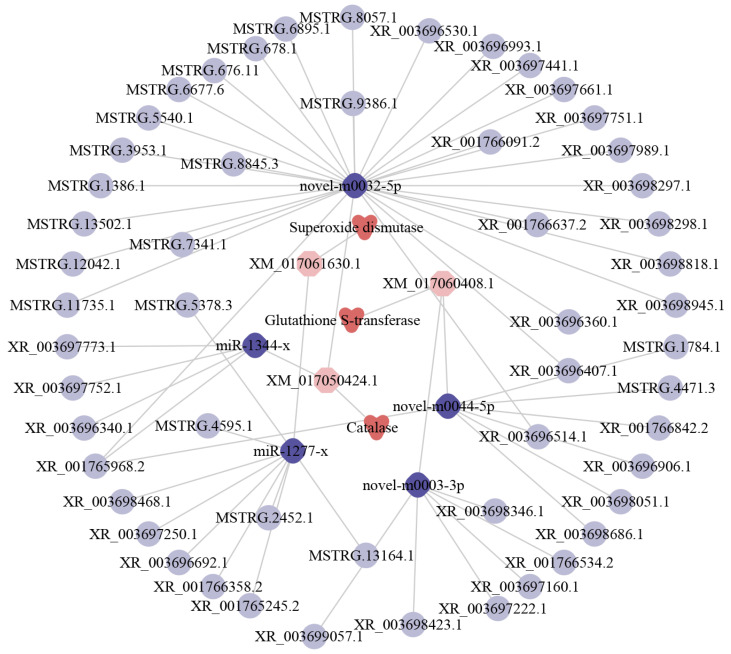
DElncRNA-DEmiRNA-DEmRNA network relative to antioxidant enzymes.

**Figure 7 ijms-24-14175-f007:**
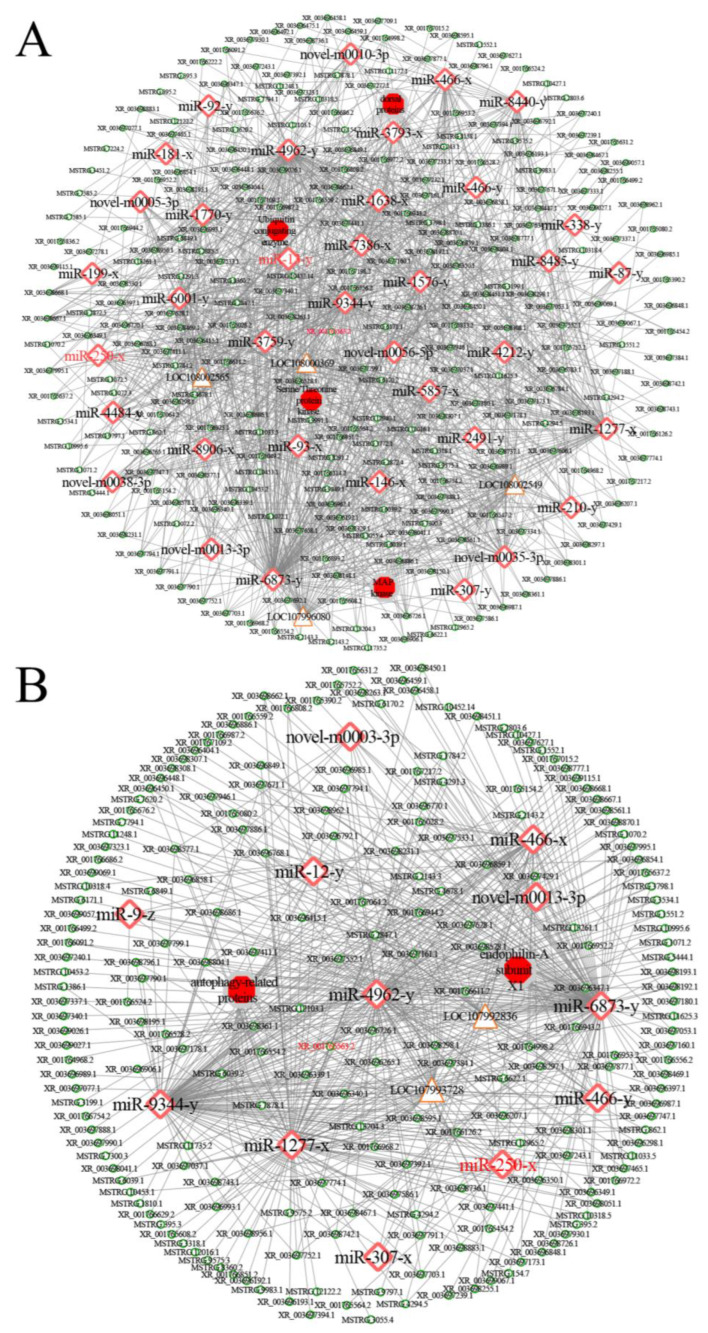
DElncRNA-involved ceRNA sub-network relative to cellular (**A**) and humoral (**B**) immune response in the *A. c. cerana* 4-day-old larval gut. Red diamonds represent miRNAs, green circles represent lncRNAs, orange triangles represent mRNAs, and red hexagons represent proteins.

**Figure 8 ijms-24-14175-f008:**
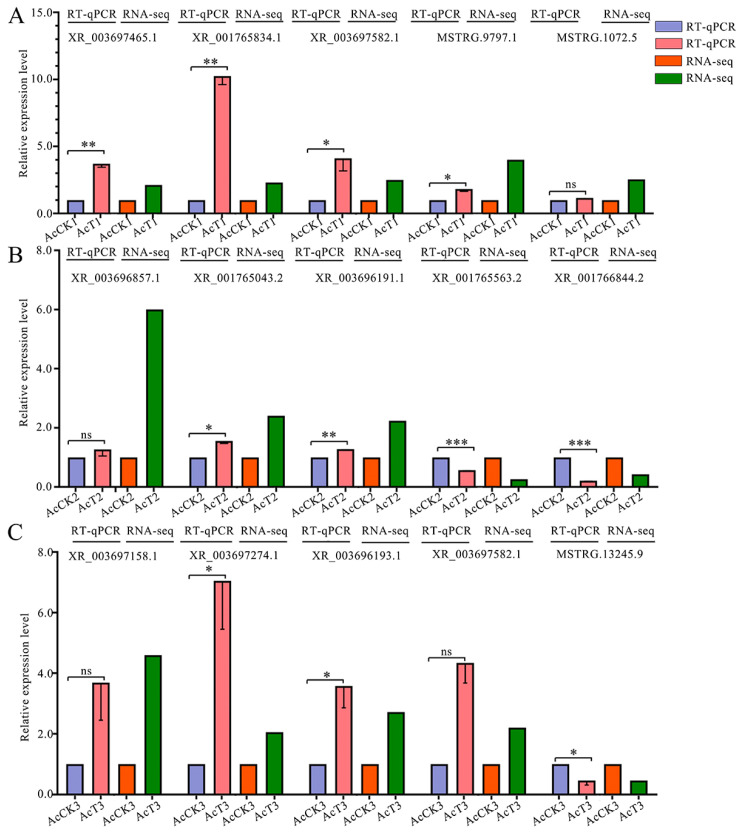
RT-qPCR detection of DElncRNAs. (**A**–**C**) Five DElncRNAs in the three comparison groups, respectively. Data were shown as Mean ± SD and subjected to Student’s *t* test (ns: *p* > 0.05, *: *p* < 0.05, **: *p* < 0.01, ***: *p* < 0.001).

**Figure 9 ijms-24-14175-f009:**
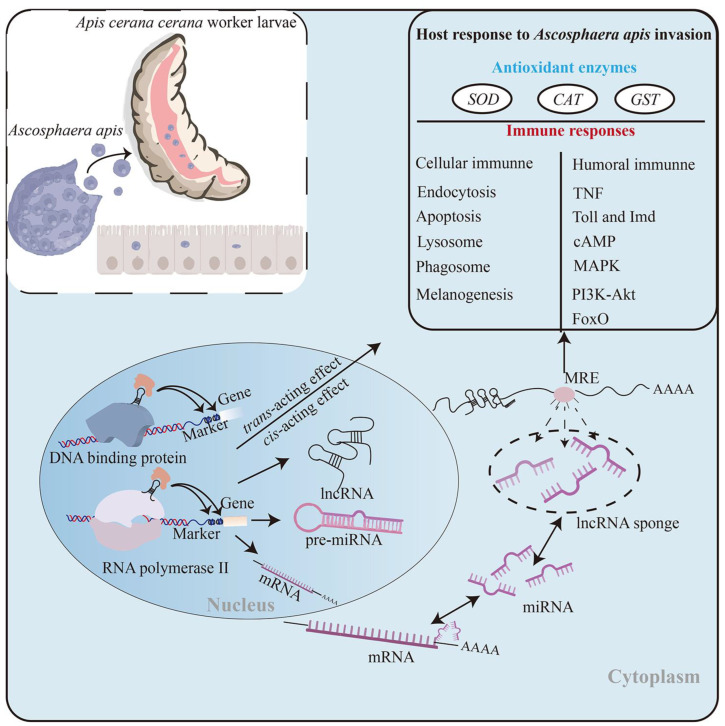
A hypothetical working model of the DElncRNA-modulated responses of *A. c. cerana* worker larvae to *A. apis* invasion.

**Table 1 ijms-24-14175-t001:** Overview of up- and down-stream genes associated with cellular and humoral immune pathways.

	Pathway	AcCK1 vs. AcT1	AcCK2 vs. AcT2	AcCK3 vs. AcT3
Cellular immune pathways	Apoptosis	1	1	0
	Apoptosis-fly	0	1	
	Phagosome	1	1	0
	Fc-γ-Receptor-Mediated Phagocytosis	1	0	0
	Autophagy-animal	1	0	0
	Melanogenesis	0	1	0
	Necroptosis	0	1	0
Humoral immune pathways	cAMP signaling pathway	1	1	0
	MAPK signaling pathway	0	1	0
	Toll and Imd signaling pathway	0	1	0
	TNF signaling pathway	0	1	0
	MAPK signaling pathway-fly	0	1	1

## Data Availability

All the data are contained within the article.

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
