# Peer review of "Regulatory Roles of Long Non-Coding RNAs Relevant to Antioxidant Enzymes and Immune Responses of Apis cerana Larvae Following Ascosphaera apis Invasion"

_ijms, 2023, doi:10.3390/ijms241814175_

Round 1

Reviewer 1 Report

Reviewer’s comments#

Manuscript describes “Regulatory roles of lncRNAs relevant to antioxidant enzymes and immune responses of Apis cerana larvae following Ascosphaera apis invasion”. The author tries to understand the role of lncRNAs on fungal infection. Techniques used in the manuscript are up to date and the experiments are well performed. Overall, the manuscript has interesting  approaches. However, I do suggest certain things which need clarification to support and strengthen the conclusion.

Abstract need to be reformatted. Please make it short.

Introduction needs improvement and elaboration related to the topic that you have chosen. Please include latest updated references. The second paragraph of introduction needs re-framing and make it simple.

“[Error! Reference source not found. - 6]” in the introduction. What do you mean by this?

Material and methods: Section 2.1 needs some corrections. Please provide some details about the fungus and the insect.

“In our previous study, the gut tissues of A. apis-inoculated A. c. cerana worker 4-, 5-, 138 and 6-day-old larvae and corresponding un-inoculated 4-, 5-, and 6-day-old larval gut 139 tissues were prepared and subjected to RNA extraction and cDNA library construction, 140 followed by transcriptome sequencing on an Illumina HiSeq 4000 platform [33]”. Please make this sentence clear?

Results: Fig. 6 is not readable. If possible, make it clear and then shift it to supplementary data. The same thing with Fig.7 and 8.

The discussion part is weak in the present form and the references are old. I recommended you to make it simple by adding the latest references and correlate with the obtained results.

General comments:

1.     In certain places the sentences need re-framing especially in case of abstract, introduction Materials and methods and discussion.

2.     Please provide all the details with proper connections between the experiments.

3.     Please provide a conclusion paragraph indicating the significance and future prospective of the findings.

4.     I do find typos throughout the manuscript and recommend the author for English correction with a native speaker.

The manuscript contains interesting findings. I ask the authors to rearrange and extend information on certain parts of the manuscript especially about the discussion and introduction with additional references. I do think that the manuscript contains important issues, information, which can lead to proper understanding the regulatory role of lncRNAs after fungal invasion. So, I consider this manuscript suitable for the publication after the suggested clarifications in the IJMS.

Recommend the author for English correction with a native speaker or through a company.

Author Response

Dear Reviewers,

Thanks so much for your comments and recommendations of great importance, which significantly improved the quality of our work. We seriously examined the whole manuscript and made corresponding modifications and improvements. All changes have been shown in red in the revised manuscript.

Response to Reviewer 1:

Major concerns:

  1. Abstract need to be reformatted. Please make it short.

Response: According to your kind comment, we seriously modified and refined the summary in the revised manuscript.

  1. Introduction needs improvement and elaboration related to the topic that you have chosen. Please include latest updated references. The second paragraph of introduction needs re-framing and make it simple.

Response: Thanks for your helpful comment, based on which we carefully modified the introduction section and added some latest related references.

  1. “[Error! Reference source not found. - 6]” in the introduction. What do you mean by this?

Response: This a mistake here due to the problem of formatting when preparing the original manuscript. We made correction in the revised version of manuscript.

  1. Material and methods: Section 2.1 needs some corrections. Please provide some details about the fungus and the insect.

Response: Following your helpful comment, we seriously examined the 2.1 section in the revised manuscript and offered more detailed information about the biomaterials used in this work. Thanks.

  1. “In our previous study, the gut tissues of A. apis-inoculated A. c. cerana worker 4-, 5-, 138 and 6-day-old larvae and corresponding un-inoculated 4-, 5-, and 6-day-old larval gut 139 tissues were prepared and subjected to RNA extraction and cDNA library construction, 140 followed by transcriptome sequencing on an Illumina HiSeq 4000 platform [33]”. Please make this sentence clear?

Response: We refined this sentence in the revised manuscript according to your kind comment.

  1. Results: Fig. 6 is not readable. If possible, make it clear and then shift it to supplementary data. The same thing with Fig.7 and 8.

Response: Thanks for your valuable recommendations. We seriously improved the quality of several figures showing regulatory networks by . Figure 8, and Figure 9 are more clear than original versions, and the original Figure 6 was displayed as a supplementary figure in the revised manuscript in that it’s very complex.

  1. The discussion part is weak in the present form and the references are old. I recommended you to make it simple by adding the latest references and correlate with the obtained results.

Response: On basis of your helpful comment and suggestion, we seriously rewritten and refined the discussion section and added some latest references in the revised manuscript. Thanks.

  1. In certain places the sentences need re-framing especially in case of abstract, introduction Materials and methods and discussion.

Response: Following your kind comment, we carefully checked the whole manuscript and made necessary modifications in the revised version of manuscript.

  1. Please provide all the details with proper connections between the experiments.

Response: Thanks for your valuable recommendation, following which we seriously checked and improved the whole materials and methods section in the original manuscript and provided necessary details of all experiments conducted in our study. Please see the revised version of manuscript for detailed information.

  1. Please provide a conclusion paragraph indicating the significance and future prospective of the findings.

Response: A conclusion paragraph was added into the revised manuscript to your kind recommendation.

  1. Ido find typos throughout the manuscript and recommend the author for English correction with a native speaker.

Response: Following your kind comment and advice, we tried our best to improve the language of the manuscript, followed by language polish by the service provided by MDPI. Thanks.

Reviewer 2 Report

I found the topic of the paper is very interesting. The text is well organized and well written and I encourage its acceptance for publication. I do not see much problems there. 

Some minor corrections and suggestions can be found in the attached file

English is fine.  Minor corrections required (mostly technical errors).

Author Response

Dear Reviewers,

Thanks so much for your comments and recommendations of great importance, which significantly improved the quality of our work. We seriously examined the whole manuscript and made corresponding modifications and improvements. All changes have been shown in red in the revised manuscript.

Response to Reviewer 2:

  1. Some minor corrections and suggestions can be found in the attached file

Response: Thank you so much for your kind comments and helpful suggestions, based on which we seriously examined the whole manuscript and made corresponding modifications in the revised manuscript.

Round 2

Reviewer 1 Report

Reviewer #:

Now, this is a very well-conceived and written paper.

The author incorporated the particulars in the present revised version of the manuscript.

Please revise the incorporation perfectly. I still found some typos which needs correction.

Once everything has been included, carefully review the references to ensure that they are all in accordance with the journal's format.

I agree that this article should be published in IJMS after such adjustments.